

# Many-body perturbation theory for strongly correlated effective Hamiltonians using effective field theory methods

**Raphaël Photopoulos[1,2,*,†] and Antoine Boulet[1,‡]**

**1** Institut Supérieur des Matériaux du Mans, ISMANS CESI École d'ingénieurs, office 51, 44 Avenue Frédéric Auguste Bartholdi, 72000 Le Mans, France
**2** IUT Grand Ouest Normandie, Université de Caen Normandie, Normandie Univ, 14000 Caen, France

★ rphotopoulos@ismans.cesi.fr ,   † raphael.photopoulos@unicaen.fr ,   ‡ aboulet@ismans.cesi.fr

## Abstract

Introducing low-energy effective Hamiltonians is usual to grasp most correlations in quantum many-body problems. For instance, such effective Hamiltonians can be treated at the mean-field level to reproduce some physical properties of interest. Employing effective Hamiltonians that contain many-body correlations renders the use of perturbative many-body techniques difficult because of the overcounting of correlations. In this work, we develop a strategy to apply an extension of the many-body perturbation theory starting from an effective interaction that contains correlations beyond the mean-field level. The goal is to re-organize the many-body calculation to avoid the overcounting of correlations originating from the introduction of correlated effective Hamiltonians in the description. For this purpose, we generalize the formulation of the Rayleigh-Schrödinger perturbation theory by including free parameters adjusted to reproduce the appropriate limits. In particular, the expansion in the bare weak-coupling regime and the strong-coupling limit serves as a valuable input to fix the value of the free parameters appearing in the resulting expression. This method avoids double counting of correlations using beyond-mean-field strategies for the description of many-body systems. The ground state energy of various systems relevant for ultracold atomic, nuclear, and condensed matter physics is reproduced qualitatively beyond the domain of validity of the standard many-body perturbation theory. Finally, our method suggests interpreting the formal results obtained as an effective field theory using the proposed reorganization of the many-body calculation. The results, like ground state energies, are improved systematically by considering higher orders in the extended many-body perturbation theory while maintaining a straightforward polynomial expansion.

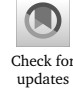

# 1  Introduction

Quantum many-body problems are, most of the time, unsolvable exactly, and several approximations have been developed to grasp the properties of such complex systems. One can mention two types of approaches that aim to solve many-body problems under some well-controlled approximations: the non-perturbative methods such as Quantum Monte Carlo (QMC) [1–6] or Density Matrix Renormalization Group (DMRG) [7–10] and the perturbative techniques such as Many-Body Perturbation Theory (MBPT) [11–17] or Coupled-Cluster (CC) theory [18–22]. On the one hand, starting from the bare Hamiltonian of the theory, non-perturbative methods can express the problem with multidimensional integrals in the Feynman's path integral formalism, which are manageable by Monte Carlo algorithms [23–26] or, alternatively, the problem can be tackled by optimizing a matrix product state tensor network using an iterative eigensolver such as the Lanczos algorithm [27,28]. On the other hand, perturbative techniques rely on a truncation scheme up to an arbitrary order of the exact solution expressed as an infinite series [29,30]. In particular, truncation at the leading order, i.e. at the first order in perturbation, is the so-called mean-field approximation.

In both perturbative and non-perturbative approaches, the description includes Beyond-Mean-Field (BMF) correlations. Another approach, namely the Density Functional Theory,[1] is a third class of methods used to solve quantum many-body problems involving BMF correlations in an effective Hamiltonian formally treated with the mean-field approximation [31–34].

---

[1]Also called Energy Density Functional (EDF) theory in the nuclear physics context.

Difficulties arise if we apply perturbative methods beyond the mean-field level from an effective Hamiltonian that contains such BMF correlations by construction. Considering an effective Hamiltonian and employing many-body methods will include BMF correlations already accounted for in the effective Hamiltonian. Several methods have been developed to avoid this overcounting in many-body techniques to deal with effective interactions at the BMF level [35]. It seems like the effective interaction to consider depends on the order of perturbation considered. One possibility to avoid overcounting of correlations is to adjust the parameter of the effective Hamiltonian according to the truncation order considered to reproduce physical properties [36]. Another strategy consists of adding counter terms in the effective interaction to avoid double counting of correlations [37].

Starting from this observation, namely the dependence of the effective interaction as a function of the perturbative order considered in the many-body technique employed, the goal of this paper is to illustrate:

1. *How to incorporate properly beyond mean-field correlations into an effective interaction, e.g. using limiting constraints, symmetries, etc.?*

2. *How to go beyond mean-field calculations using effective interactions that already contain such correlations?*

We restrict our development to the case of the MBPT to clarify the formal link between the effective interaction and the truncation order of the perturbative method. The purpose of this work is then to develop a systematic MBPT for strongly correlated effective Hamiltonians.

The article is organized as follows. In section 2, the framework of our development is briefly presented and motivated to generalize the Rayleigh-Schrödinger perturbation theory. In section 3, we develop a reformulation of the MBPT exhibiting a new extra parameter that allows the use of an effective Hamiltonian that is the core of the presented work. We also discuss the interpretation of our extended MBPT in the Effective Field Theory (EFT) framework. Then, in sections 4, 5 and 6, we apply our strategy for several many-body problems encountered, respectively, in the context of ultracold atomic quantum gases, nuclear and condensed matter physics, as a proof of principle. Finally, in section 7, we conclude and discuss the implications and further developments of our approach for the description of quantum many-body systems.

## 2 Presentation of the problem

We consider a many-particle system of $N$ fermions interacting through a Hamiltonian parameterized by a tunable coupling constant $\lambda$ and decomposed as:

$$\hat{H}_\lambda = \hat{H}_0 + \lambda \hat{h}. \tag{1}$$

The reference state $|\Phi_0\rangle$ is chosen as the solution of the static many-body Schrödinger equation involving $\hat{H}_0$, that is, $\hat{H}_0 |\Phi_0\rangle = E_0 |\Phi_0\rangle$ with the unperturbed energy $E_0$. In this work, we only consider the normal component, making the reference state $|\Phi_0\rangle$ the Hartree-Fock Slater determinant associated with $\hat{H}_0$.

Then, we assume that we know the Taylor expansion of the Ground State (GS) energy around $\lambda = 0$ and its value at the limit $\lambda \to \infty$. These two limits, respectively $\lambda \ll 1$ and $\lambda \gg 1$, will be named respectively low- and high-scale limits in the following. For simplicity, we use dimensionless GS energy. We then write:

$$\frac{E_\lambda}{E_0} = 1 + \sum_{n=1}^{\infty} \gamma_n \lambda^n, \tag{2}$$

where we have introduced the set $\{\gamma_n\}$ for the low-scale expansion of the dimensionless GS energy. In the high-scale limit, we define the parameter $\xi_0$ as:

$$\lim_{\lambda \to \infty} \frac{E_\lambda}{E_0} = \xi_0 . \tag{3}$$

At the standard Hartree-Fock (HF) level or first-order MBPT, i.e. when $\lambda \hat{h}$ can be considered in perturbation according to $\hat{H}_0$ (low-scale limit $\lambda \ll 1$), the energy of the system is given by the parameter $\gamma_1$ only: $E_\lambda/E_0 = 1 + \gamma_1 \lambda + \mathcal{O}(\lambda^2)$. In general, the result of any order will give a polynomial expansion in $\lambda$ that diverges in the strong-coupling regime (high-scale limit $\lambda \gg 1$). Thus, in this sense, the standard MBPT does not provide a finite value of the observable in this limit. The goal of this paper is to propose a method based on the MBPT at a given order and relying on the use of an effective parameterization of the Hamiltonian such that:

*(i)* we recover the low-scale limit up to the order of the MBPT considered;

*(ii)* we impose the high-scale results $E_\infty/E_0 = \xi_0$, where the parameter $\xi_0$ is assumed to be known.

In the following, we will arbitrarily assume the particular form of the effective Hamiltonian as follows:

$$\widetilde{H}_\lambda = \hat{H}_0 + \frac{\lambda \hat{h}}{1 + \lambda \gamma_1/a} , \tag{4}$$

which mimic, in a simple manner,[2] the bare Hamiltonian (1) up to first order in power of $\lambda$, i.e. $\widetilde{H}_\lambda = \hat{H}_\lambda + \mathcal{O}(\lambda^2)$, but having a finite limit in the high-scale regime tuned by the parameter $a$. In other words, the bare coupling constant $\lambda$ is replaced by $\lambda/(1 + \lambda \gamma_1/a)$ which can be interpreted as a renormalized coupling constant due to the medium effect [38]. This extra parameterization allows one to include many-body BMF correlations, e.g. the high-scale constraint (3), directly in the effective interaction. Using this particular form of effective Hamiltonians, we will propose a systematic method to perform MBPT calculations capable of reproducing the low-scale expansion of the GS energy (2) up to a given order in perturbation and the high-scale limit (3).

In the following, we will first illustrate our motivations with a recent example of the GS energy of diluted many-body Fermi systems at the HF level using an effective Hamiltonian equivalent to (4). Then, we extend the method to higher orders in perturbation using a reformulation of the MBPT that allows flexibility to adjust some parameters on the proper low- and high-scale constraints.

## 2.1 Hartree-Fock or MBPT at first order

The HF energy using the effective Hamiltonian (4) is given by the Leading Order (LO) in perturbation:

$$E_\lambda^{LO} = E_0 + \frac{\langle \Phi_0 | \lambda \hat{h} | \Phi_0 \rangle}{1 + \lambda \gamma_1/a} . \tag{5}$$

Identifying with the low-scale expansion (2), we have $\langle \Phi_0 | \lambda \hat{h} | \Phi_0 \rangle = \gamma_1 \lambda E_0$. Then, imposing the exact high-scale limit constraint (3), we get $\xi_0 = 1 + a$. This expression of the high-scale parameter is valid only at the HF level and depends on the form chosen to define the effective Hamiltonian (4).

---

[2]In general, $\widetilde{H}_\lambda = \hat{H}_0 + F_a(\lambda)\hat{h}$ where the function $F_a$ has the following properties: $F_a(\lambda) = \lambda + \mathcal{O}(\lambda^2)$ and $F_a(\infty) = a/\gamma_1$.

## 2.2 Direct application to ultracold atom systems

In this section, as a proof of principle, we consider an infinite spin-saturated diluted system of fermions at zero temperature. In that case, the Hamiltonian is the sum of the kinetic Hamiltonian and a two-body contact interaction of the form: $\langle \mathbf{k'}|\lambda \hat{h}|\mathbf{k}\rangle = 4\pi a_s/m$ [39] where $a_s$ is the $s$-wave scattering length and plays the role of the parameter $\lambda$. Here, we use the convention $\hbar = 1$. We know that within the framework of the MBPT applied to dilute Fermi gases with an attractive interaction ($a_s < 0$), this corresponds to the Bardeen-Cooper-Schrieffer (BCS) regime [40,41]:

$$\frac{E}{E_0} = 1 + \frac{10}{9\pi}(a_s k_F) + \mathcal{O}[(a_s k_F)^2], \tag{6a}$$

$$\lim_{|a_s| \to \infty} \frac{E}{E_0} = \xi, \tag{6b}$$

where $\xi = 0.376(4)$ [42] is the so-called Bertsch parameter, $k_F$ is the Fermi momentum of the system directly linked to the density $\rho = k_F^3/3\pi^2$, and $E_0/N = 3k_F^2/10m$ is the free Fermi gas energy.

Then we can directly identify and make the correspondence $\lambda = a_s$, $\gamma_1 = 10k_F/9\pi$ and $\xi_0 = \xi$ that leads to the HF energy (5):

$$\frac{E^{LO}}{E_0} = 1 + \frac{\dfrac{10}{9\pi}(a_s k_F)}{1 + \dfrac{10}{9\pi}(\xi - 1)^{-1}(a_s k_F)}. \tag{7}$$

This expression was proposed several times during the last decade in the context of ultracold atoms using a Padé[1/1] form of the energy and the unitary limit as a constraint [43,44]. In figure 1, we display the GS energy of the spin-saturated Fermi gas at the HF level, denoted as $E^{LO}$, as a function of $(a_s k_F)$ and obtained with (7) using the correspondences of parameters for the dilute Fermi gas. As discussed in [43,44], this functional qualitatively reproduces well the GS energy and thermodynamical properties of diluted ultracold atom systems across the weak to the strong coupling regimes in a very compact form, explicitly in terms of the density and the low-energy constant of the bare two-body interaction.

Including higher-order MBPT contributions (or applying more high-scale constraints such as the Taylor expansion in the limit $\lambda \to \infty$) at the HF level is not feasible in a systematic manner. For instance, various parameterizations of the effective Hamiltonian can lead to divergences in certain cases. Such developments are not the purpose of this work. Below, we suggest pursuing BMF calculations keeping the idea and the form of the Hamiltonian discussed up to now. In the following, we develop a method that allows us to extend MBPT for an effective Hamiltonian similar to (4) and keep the low- and high-scale limits valid. The aim is to elaborate a method that avoids naturally the double counting of correlations arising from the association of perturbative techniques and effective Hamiltonians.

## 3 Extension of the MBPT with strongly correlated effective Hamiltonians

In this section, we propose an extension of the MBPT allowing the use of effective Hamiltonians (4). We first recall the equation of MBPT in Rayleigh-Schrödinger formalism and then propose another truncation scheme by introducing a free parameter. This new formulation of the theory allows for some flexibility to perform the MBPT up to a given order using the effective Hamiltonian (4) and keep the proper low-scale expansion up to the order considered, as well as the high-scale constraint.

### 3.1 Many-body perturbation theory with a bare Hamiltonian

Assuming that the GS of the Hamiltonian $\hat{H}_\lambda = \hat{H}_0 + \lambda\hat{h}$ for the many-body system of interest is nondegenerate [12,45], we decompose the exact many-body GS $|\Psi_0\rangle$ as a series of reference states, e.g. Slater determinants, as:

$$|\Psi_0\rangle = |\Phi_0\rangle + \sum_{i\neq 0} \mathcal{C}_i |\Phi_i\rangle \,, \tag{8}$$

where $|\Phi_0\rangle$ is the GS of the unperturbed system, i.e. $\hat{H}_0|\Phi_0\rangle = E_0|\Phi_0\rangle$, $E$ is the exact GS energy of the system, the set $\{|\Phi_i\rangle\}$ denotes the eigenvector basis of the unperturbed Hamiltonian $\hat{H}_0$ (note that we use the intermediate normalization $\langle\Psi_0|\Phi_0\rangle = 1$), and the $\{\mathcal{C}_i\}$ are the associated amplitudes of the state in this basis. The static many-body Schrödinger equation reads:

$$\hat{H}_\lambda|\Psi_0\rangle = E|\Psi_0\rangle \,, \tag{9}$$

and using the hermiticity of the Hamiltonian, we define the energy shift as:

$$\Delta E = E - E_0 = \langle\Phi_0|\lambda\hat{h}|\Psi_0\rangle \,. \tag{10}$$

Then, we define the projectors $\hat{P} = |\Phi_0\rangle\langle\Phi_0|$ and $\hat{Q} = \sum_{i\neq 0}|\Phi_i\rangle\langle\Phi_i|$ such that $|\Psi_0\rangle = (\hat{P}+\hat{Q})|\Psi_0\rangle = |\Phi_0\rangle + \hat{Q}|\Psi_0\rangle$. Now, starting from the Schrödinger equation (9), we introduce an energy operator $\hat{\omega}$ and we project the result into the sub-Hilbert space defined by $\hat{Q}$ to get:

$$|\Psi_0\rangle = |\Phi_0\rangle + \hat{R}(\hat{\omega})\cdot(\hat{\omega} - E + \lambda\hat{h})|\Psi_0\rangle \,, \tag{11}$$

where we denote the resolvent operator (that is assumed to exist) by:

$$\hat{R}(\hat{\omega}) = \frac{\hat{Q}}{\hat{\omega} - \hat{H}_0} \,. \tag{12}$$

By consecutive iterations, we can write the energy shift (10) and the exact many-body state (11) as a perturbative series:

$$\Delta E = \sum_{n=1}^{\infty} \langle\Phi_0|\,\lambda\hat{h}\cdot[\hat{R}(\hat{\omega})\cdot(\hat{\omega} - E + \lambda\hat{h})]^{n-1}|\Phi_0\rangle \,, \tag{13a}$$

$$|\Psi_0\rangle = \sum_{n=1}^{\infty} [\hat{R}(\hat{\omega})\cdot(\hat{\omega} - E + \lambda\hat{h})]^{n-1}|\Phi_0\rangle \,. \tag{13b}$$

#### 3.1.1 Rayleigh-Schrödinger perturbation theory

In Rayleigh-Schrödinger Perturbation Theory (RSPT), we chose $\hat{\omega} = E_0$. Using that particular choice, the energy shift (13a) and the exact many-body state (13b) can be written as a perturbative series:

$$\Delta E = \sum_{n=1}^{\infty} \langle\Phi_0|\,\lambda\hat{h}\cdot[\hat{R}(E_0)\cdot(\lambda\hat{h} - \Delta E)]^{n-1}|\Phi_0\rangle \,, \tag{14a}$$

$$|\Psi_0\rangle = \sum_{n=1}^{\infty} [\hat{R}(E_0)\cdot(\lambda\hat{h} - \Delta E)]^{n-1}|\Phi_0\rangle \,. \tag{14b}$$

Then, expanding the energy shift as a series $\Delta E = \sum_n \Delta E^{(n)}$ in powers of $\lambda$ and using the fact that $\hat{Q}\Delta E |\Phi_0\rangle = 0$, we can get the expression of $\Delta E$ as a series in terms of the interaction strength $\lambda$. For instance, up to the third order in power of $\lambda$:

$$\Delta E^{(1)} = \langle \Phi_0 | \lambda \hat{h} | \Phi_0 \rangle \,, \tag{15a}$$

$$\Delta E^{(2)} = \langle \Phi_0 | \lambda \hat{h} \cdot \hat{R}_0 \cdot \lambda \hat{h} | \Phi_0 \rangle \,, \tag{15b}$$

$$\Delta E^{(3)} = \langle \Phi_0 | \lambda \hat{h} \cdot \hat{R}_0 \cdot \lambda \hat{w} \cdot \hat{R}_0 \cdot \lambda \hat{h} | \Phi_0 \rangle \,, \tag{15c}$$

where we have defined $\hat{R}_0 = \hat{R}(E_0)$ and $\lambda \hat{w} = \lambda \hat{h} - \Delta E^{(1)}$. We can remark that by matching the low-scale expansion for the energy in series of $\lambda^n$ given by (2), we have, by definition, $\Delta E^{(n)} = \gamma_n \lambda^n E_0$.

### 3.1.2 Alternative choice for the resolvent parameter

The choice (12) for the resolvent operator is not unique. For example, the Brillouin-Wigner perturbation theory consists of setting $\hat{\omega} = E$.

In this work, we propose to set $\hat{\omega} = \hat{H}_0 + \beta(E_0 - \hat{H}_0) = \hat{\omega}_\beta$, such that:

$$\hat{R}(\hat{\omega}_\beta) = \frac{1}{\beta} \frac{\hat{Q}}{E_0 - \hat{H}_0} = \frac{1}{\beta} \hat{R}_0 \,, \tag{16}$$

where $\beta$ is a number to be determined. Note that the case $\beta = 1$ is equivalent to RSPT. The energy shift (13a) and the exact many-body state (13b) are now given by:

$$\Delta E(\beta) = \sum_{n=1}^{\infty} \langle \Phi_0 | \lambda \hat{h} \cdot \hat{B}_n(\beta) | \Phi_0 \rangle \,, \tag{17a}$$

$$|\Psi_0(\beta)\rangle = \sum_{n=1}^{\infty} \hat{B}_n(\beta) |\Phi_0\rangle \,, \tag{17b}$$

where we have defined $\hat{B}_1(\beta) = 1$ and for $n \geq 2$:

$$\hat{B}_n(\beta) = \frac{1}{\beta^{n-1}} [(\beta - 1) + \hat{R}_0 \cdot (\lambda \hat{h} - \Delta E)]^{n-2} \cdot \hat{R}_0 \cdot \lambda \hat{h} \,. \tag{17c}$$

To obtain these expressions, we use $(\hat{H}_0 - E_0)|\Phi_0\rangle = 0$, $\hat{Q}\Delta E |\Phi_0\rangle = 0$, the relationship $\hat{R}_0 \cdot (E_0 - \hat{H}_0) \cdot \hat{R}_0 = \hat{R}_0$ since $\hat{Q}$ is idempotent ($\hat{Q}^2 = \hat{Q}$), and the commutation property of $\hat{H}_0$ with $\hat{R}_0$.

We can show that, following the same truncation scheme in terms of the power of $\lambda$ as in RSPT, each term is re-summed exactly to be $\beta$-independent at each order and consequently $\Delta E(\beta) = \Delta E$. More precisely, we have:

$$\sum_{n=2}^{\infty} \hat{B}_n(\beta) = \frac{1}{1 - \hat{R}_0 \cdot (\lambda \hat{h} - \Delta E)} \cdot \hat{R}_0 \cdot \lambda \hat{h}$$

$$= \sum_{n=2}^{\infty} [\hat{R}_0 \cdot (\lambda \hat{h} - \Delta E)]^{n-2} \cdot \hat{R}_0 \cdot \lambda \hat{h} \,,$$

and we then recover the RSPT results (14), i.e. the $\beta$-independent results.

Considering the bare interaction Hamiltonian $\lambda \hat{h}$, the only (or more natural) possibility is to use the RSPT formalism, that is to set $\beta = 1$ or to use the simplified $\beta$-independent result above, which appears naturally in calculations. However, using an effective Hamiltonian like (4), the flexibility of the $\beta$ parameter can be used to match the low-scale expansion at a given order in MBPT. In the following, we illustrate this aspect at the first, second, and third orders using the effective Hamiltonian (4).

## 3.2 Extended many-body perturbation theory with an effective Hamiltonian

We consider the effective Hamiltonian (4) where the parameter $a$ is to be determined by containing the GS energy[3] at $N^l$LO on the high-scale limit constraint (3). Using the preliminary result on the MBPT scheme proposed in section 3.1.2, the energy shift (13a) and the exact many-body state (13b) are now given by:

$$\Delta E_\lambda(\beta) = \sum_{n=1}^{\infty} \frac{\langle \Phi_0 | \lambda \hat{h} \cdot \widetilde{B}_n(\beta) | \Phi_0 \rangle}{1 + \lambda \gamma_1/a}, \tag{18a}$$

$$|\Psi_0(\beta)\rangle = \sum_{n=1}^{\infty} \widetilde{B}_n(\beta) |\Phi_0\rangle, \tag{18b}$$

where we have defined $\widetilde{B}_1(\beta) = 1$ and for $n \geq 2$:

$$\widetilde{B}_n(\beta) = \frac{1}{\beta^{n-1}} \left[ (\beta - 1) + \hat{R}_0 \cdot \left( \frac{\lambda \hat{h}}{1 + \lambda \gamma_1/a} - \Delta E_\lambda(\beta) \right) \right]^{n-2} \cdot \hat{R}_0 \cdot \frac{\lambda \hat{h}}{1 + \lambda \gamma_1/a}. \tag{18c}$$

The basic idea of performing MBPT calculations using the particular choice (4) for an effective Hamiltonian is to reorganize the standard truncation scheme and adjust the $\beta$ parameter to match the low-scale MBPT expansion. More precisely, we will take advantage of the reformulation of the RSPT to express the GS energy and the many-body state at $N^l$LO as series of the form:

$$\frac{E_\lambda^{N^l LO}}{E_0} = 1 + \sum_{n=1}^{l+1} \left( \frac{\lambda}{f_\lambda^{N^l LO}} \right)^n \Gamma_n^{N^l LO}, \tag{19a}$$

$$|\Psi_0^{N^l LO}\rangle = \sum_{n=1}^{l+1} \left( \frac{\lambda}{f_\lambda^{N^l LO}} \right)^{n-1} \hat{G}_n^{N^l LO} |\Phi_0\rangle, \tag{19b}$$

where $f_\lambda^{N^l LO} = 1 + \lambda \gamma_1/a$. In the following, we illustrate the strategy in the first ($l = 0$), second ($l = 1$), and third ($l = 2$) order, and the parameter $a$ will now be denoted $a_{l+1}$ to avoid confusion.

### 3.2.1 First order

In that case, the GS energy is given by the energy shift (18) truncated at $l = 0$, that is:

$$\frac{E_\lambda^{LO}}{E_0} = 1 + \gamma_1 \frac{\lambda}{f_\lambda^{LO}}, \tag{20}$$

where we can identify $\Gamma_1^{LO} = \gamma_1$ using the fact that $\langle \Phi_0 | \lambda \hat{h} | \Phi_0 \rangle = \gamma_1 \lambda E_0$. It remains the determination of the high-scale parameter using (3). As above, this gives $\xi_0 = 1 + a_1$.

### 3.2.2 Second order

Following the same strategy, we truncate (18) to $l = 1$. We have first to determine:

$$\begin{aligned}
\frac{\langle \Phi_0 | \lambda \hat{h} \cdot \widetilde{B}_2(\beta) | \Phi_0 \rangle}{1 + \lambda \gamma_1/a_2} &= \frac{1}{\beta} \frac{\langle \Phi_0 | \lambda \hat{h} \cdot \hat{R}_0 \cdot \lambda \hat{h} | \Phi_0 \rangle}{(1 + \lambda \gamma_1/a_2)^2} \\
&= \frac{\gamma_2}{\beta} \left( \frac{\lambda}{1 + \lambda \gamma_1/a_2} \right)^2 E_0.
\end{aligned} \tag{21}$$

---

[3]The notation $N^l$LO refer to the next-to-next-...-to-next leading order. For example, the next-to-leading order (NLO) corresponds to the second order in perturbation, the next-to-next-to-leading-order ($N^2$LO) corresponds to the third order in perturbation, etc.

Then, the GS energy at NLO reads:

$$\frac{E_\lambda^{NLO}}{E_0} = 1 + \gamma_1 \frac{\lambda}{f_\lambda^{NLO}} + \frac{\gamma_2}{\beta}\left(\frac{\lambda}{f_\lambda^{NLO}}\right)^2.$$ (22)

**Low-scale expansion**   Now, we determine the $\beta$ parameter to match the low-scale expansion up to the second order, i.e. such that:

$$\frac{E_\lambda^{NLO}}{E_0} = 1 + \gamma_1\lambda + \gamma_2\lambda^2 + \mathcal{O}(\lambda^3),$$ (23)

leading to:

$$\frac{1}{\beta} = 1 + \frac{\gamma_1^2}{a_2\gamma_2}.$$ (24)

**High-scale constraint**   Finally, we fix the high-scale parameter $a_2$ using (3) leading to the quadratic equation:

$$\xi_0 = 1 + 2a_2 + \frac{\gamma_2}{\gamma_1^2}a_2^2.$$ (25)

This quadratic equation admits two independent solutions for $a_2$. The choice of solutions to consider is discussed in the latter.

**Many-body state**   Just above, we have fixed all the parameters used in the method proposed by matching on the low-scale expansion (2) and on the high-scale constraint (3). Thus, we are now able to express the many-body state in that new formulation of the MBPT using (18). To be more explicit, $\Gamma_n^{NLO}$ and $\hat{G}_n^{NLO}$ appearing in (19) are given by $\Gamma_1^{NLO} = \gamma_1$, $\hat{G}_1^{NLO} = 1$ and:

$$\Gamma_2^{NLO} = \left[1 + \frac{\gamma_1^2}{a_2\gamma_2}\right]\gamma_2,$$

$$\hat{G}_2^{NLO} = \left[1 + \frac{\gamma_1^2}{a_2\gamma_2}\right]\hat{R}_0 \cdot \hat{h}.$$

### 3.2.3   Third order

In this part, we illustrate the truncation method that we propose at the third order ($l = 2$) in perturbation. For that, we use the flexibility to adjust the $\beta$ parameter *a posteriori* on the low-scale expansion and the property that, as discussed before, the energy shift (17) is, in fact, $\beta$-independent. At the second order in perturbation (see above), only one $\beta$ parameter is required to obtain the proper low-scale expansion. But, in the third order, we need two different $\beta$ parameters to match the low-scale expansion up to the third order. Here, the strategy consists of including an additional parameter to (18) as:

$$\Delta E_\lambda(\beta) \rightarrow \frac{\Delta E_\lambda(\beta_1) + \Delta E_\lambda(\beta_2)}{2}.$$ (26)

We note that this choice is arbitrary in the sense that any weighted arithmetic mean of the form $\Delta E(\beta) \rightarrow r\Delta E(\beta_1) + (1-r)\Delta E(\beta_2)$ can be used, where $0 < r < 1$ is a parameter to adjust. For simplicity and because the strategy developed in this work implies to adjust only three parameters at the third order in perturbation on the low- and high-scale limits, we chose

the arithmetic mean $r = 1/2$. Then, making a truncation of (18) up to $l = 2$, we have the following:

$$
\begin{aligned}
\frac{\langle \Phi_0 | \lambda \hat{h} \cdot \widetilde{B}_3(\beta) | \Phi_0 \rangle}{1 + \lambda \gamma_1 / a_3} &= \frac{\beta - 1}{\beta^2} \frac{\langle \Phi_0 | \lambda \hat{h} \cdot \hat{R}_0 \cdot \lambda \hat{h} | \Phi_0 \rangle}{(1 + \lambda \gamma_1 / a_3)^2} + \frac{1}{\beta^2} \frac{\langle \Phi_0 | \lambda \hat{h} \cdot \hat{R}_0 \cdot \lambda \hat{w} \cdot \hat{R}_0 \cdot \lambda \hat{h} | \Phi_0 \rangle}{(1 + \lambda \gamma_1 / a_3)^3} \\
&= \frac{(\beta - 1) \gamma_2}{\beta^2} \left( \frac{\lambda}{1 + \lambda \gamma_1 / a_3} \right)^2 E_0 + \frac{\gamma_3}{\beta^2} \left( \frac{\lambda}{1 + \lambda \gamma_1 / a_3} \right)^3 E_0 ,
\end{aligned}
\tag{27}
$$

where we have moreover truncated the expression of $\widetilde{B}_3(\beta)$ up to the third order in power of $\lambda$ (as in the standard RSPT formulation). Finally, using (26), the N$^2$LO GS energy is given by:

$$
\frac{E_\lambda^{N^2LO}}{E_0} = 1 + \gamma_1 \frac{\lambda}{f_\lambda^{N^2LO}} + \frac{\gamma_2}{2} \left[ \frac{2\beta_1 - 1}{\beta_1^2} + \frac{2\beta_2 - 1}{\beta_2^2} \right] \left( \frac{\lambda}{f_\lambda^{N^2LO}} \right)^2 + \frac{\gamma_3}{2} \left[ \frac{1}{\beta_1^2} + \frac{1}{\beta_2^2} \right] \left( \frac{\lambda}{f_\lambda^{N^2LO}} \right)^3 .
\tag{28}
$$

As before, the parameters $\{\beta_1, \beta_2\}$ are to be determined in terms of $\{\gamma_1, \gamma_2, \gamma_3\}$ and $a_3$ to match the low-scale expansion up to the third order, and then the high-scale parameter $a_3$ can be determined by imposing the high-scale constraint for the GS energy.

### 3.2.4 Avoiding the double counting of correlations

We can observe that our method does not overcount the many-body correlations. This is due to the fact that we impose the low-scale expansion up to a given order in perturbation. Thus, even if many-body correlations are included in an effective Hamiltonian and potentially taken into account within the MBPT framework, the adjustment of the $\beta$ parameters in order to reproduce the low-scale expansion (2) in the limit $\lambda \ll 1$ avoid this eventual double counting. To be more precise, if we reduce our strategy to the RSPT, that is to say setting $\beta = 1$, and keeping an effective Hamiltonian similar to (4), the low-scale expansion (2) cannot be recovered in the limit $\lambda \ll 1$. Therefore, our strategy consists in mimic the infinite series (2) with effective parameters $\{\widetilde{\gamma}_n\}$ for which the first parameters match the physical parameters $\{\gamma_n\}$ up to the order $m$ that is considered to truncate the MBPT calculation, i.e. $\widetilde{\gamma}_1 = \gamma_1, \ldots, \widetilde{\gamma}_m = \gamma_m$, and $\widetilde{\gamma}_n \neq \gamma_n$ for $n > m$.

### 3.3 Effective field theory interpretation of the extended MBPT

Considering the new method proposed and described above using a correlated effective Hamiltonian, a general and comprehensive Effective Field Theory (EFT) expansion emerges. Indeed, in the GS energy expansion (19) obtained, $\lambda$ plays the role of the low-scale and $f_\lambda^{N^lLO}$ the role of the high-scale. This expansion is similar to the low-scale expansion (2) except that the low-scale parameter $\lambda$ is renormalized by the high-scale in-medium function $f_\lambda^{N^lLO}$. The proposed method is a systematic procedure to:

- get a better convergence of the result compared to a standard MBPT approach, e.g. finite limit in the strong coupling regime (except when $f_\lambda^{N^lLO} = 0$);

- include correlations in an effective way much BMF correlations and even much beyond the correlations accessible using the standard MBPT approach;

- allow the use of the MBPT systematically using an effective correlated Hamiltonian (4);

- reorganize the MBPT expansion to be valid in the low- and high-scale limit;

- remove automatically the double counting of correlations included in the effective Hamiltonian (4) and resulting of the MBPT approach;

- obtain an EFT expansion of the observables in term of $(\lambda/f_\lambda^{N^l LO})^n$ where $\lambda$ and $f_\lambda^{N^l LO}$ play the role of the low- and high-scales, respectively;

- have analytical, simple, and compact expressions for the observable as a function of well-determined fixed quantities, e.g. $\{\gamma_n\}$ and $\xi_0$, without fitting procedure (eventually the used coefficients can be provided by *ab initio* calculations or experiments).

But two problems remain:

*(i)* The polynomial equation similar to (25) admits $l$ different complex solutions for the determination of the parameter $a_{l+1}$ defining the effective Hamiltonian (4).

*(ii)* The GS has an imaginary non-physical part due to the fact that we allow a complex value for the $a_{l+1}$ parameter.

Before discussing the results for some applications of the extended MBPT method developed and applied to physical systems, we give a simple argument and an explanation to *(i)* make a pragmatic choice for the parameter $a_{l+1}$ to use and *(ii)* remove the pathology and recover a real GS energy.

### 3.3.1 Choice of the high-scale solution

The $N^l$LO, the GS energy in the limit $\lambda \to \infty$ is given by a polynomial equation similar to (25). The main purpose of the proposed method is to grasp the high-scale correlations from the HF level. Thus, this motivates the choice of the minimum solution, i.e. set $a_{l+1} = \min\{|x_{l+1}|\}$, where the set $\{x_{l+1}\}$ denotes the $l$ solutions of the high-scale polynomial equation similar to (25), such that the main contribution to the energy in the high-scale limit is given by the leading order[4] of the expansion (19).

### 3.3.2 Remove the imaginary part

The appearance of an imaginary part in the GS energy comes from the resolution of the polynomial equation similar to (25) having $l+1$ distinct solutions. However, the low-scale expansion of the GS energy (19) matches the standard MBPT expansion up to the $l+1$ order and the high-scale constraint (3). Consequently, we have the following properties:

$$\text{Im}\left(\frac{E_\lambda^{N^l LO}}{E_0}\right) = \mathcal{O}(\lambda^{l+2}).$$

This flexibility allows the redefinition of $\Gamma_{l+1}^{N^l LO}$ to get a real GS energy since this change does not affect the systematic procedure requiring only the first $l+1$ order of the low-scale expansion. Thus, this minimal subtraction of the imaginary part consists of keeping the real part of the GS energy.

## 4 Quantum gases context: Application to ultracold atom systems

As in section 2.2, the correspondence for spin-saturated dilute ultracold atom systems with the $N^l$LO GS energy is given by the high-scale constraint, i.e. the Bertsch parameter $\xi_0 = 0.376$, and the low-scale expansion (2) where $\gamma_1 = 10k_F/9\pi$, $\gamma_2 = 4(11-2\ln 2)k_F^2/21\pi^2$, $\gamma_3 = 0.032k_F^3$, and $\gamma_4 = 0.451k_F^4$ [46]. These results are summarized in table 1.

---

[4]Valid in the case $|a_{l+1}| < 1$, but the same conclusion on this choice occurs to minimize the correction of the next order.

Table 1: Low-scale coefficients and high-scale limit (Bertsch parameter), for $(a_s k_F) > 0$ and $(a_s k_F) < 0$, of the ultracold Fermi gas. The dashes mean that the coefficient is not defined.

|  | $(a_s k_F) > 0$ | $(a_s k_F) < 0$ |
|---|---|---|
| $\gamma_1$ | $k_F/6\pi$ | $10 k_F/9\pi$ |
| $\gamma_2$ | – | $4 k_F^2(11 - 2\ln 2)/21\pi^2$ |
| $\gamma_3$ | – | $0.032 k_F^3$ |
| $\xi_0$ | 0.376 | 0.376 |

We show in figure 1 the GS energy of the spin-saturated Fermi gas at N$^l$LO, $E^{N^l LO}$ given by (19), as a function of $(a_s k_F)$. On the BCS side of the crossover ($a_s < 0$), the results obtained with our method go much beyond the standard MBPT approach and reproduce very well the *ab initio* calculations from the weak to the strong coupling regime without substantial improvement when we increase the order of perturbation using the effective Hamiltonian (4). On the Bose-Einstein Condensate (BEC) side of the crossover ($a_s > 0$), due to the appearance of bound dimers in the gas, the GS energy of the system (containing the binding energies of the dimers) takes the following expansion [43, 47]:

$$\frac{E}{E_0} = \frac{1}{6\pi}(a_s k_F) + \frac{32}{375\sqrt{\pi^5/10}}(a_s k_F)^{5/2} + \cdots . \tag{29}$$

This expansion does not exhibit a simple polynomial form in $(a_s k_F)$ and, thus, the proposed method, in particular the choice of the effective Hamiltonian (4), is not adapted for such a low-scale expansion (except up to the first order). That is why we do not discuss this case further. For reference, the leading order is given in figure 1(c,d) and shows a large error in the non-perturbative regime ($a_s k_F > 1$).

## 5 Nuclear physics context: Application to the Richardson pairing Hamiltonian

We consider a quantum many-body system composed of $N$ doubly degenerate in spin $\sigma = \uparrow, \downarrow$ and equally spaced single-particle levels $n = 1, 2, \ldots$. The many-body Hamiltonian expressed in the second quantization formalism, with at most two-body interaction, is given by the so-called Richardson Pairing Hamiltonian[5] (RPH) [50]:

$$\begin{aligned} \hat{H}_{RPH} &= \hat{H}_0 + \hat{V} \\ &= e \sum_{n\sigma}(n-1)\hat{a}_{n\sigma}^\dagger \hat{a}_{n\sigma} - \frac{g}{2}\sum_{nm}\hat{a}_{n\uparrow}^\dagger \hat{a}_{n\downarrow}^\dagger \hat{a}_{m\downarrow}\hat{a}_{m\uparrow} . \end{aligned} \tag{30}$$

For simplicity and without loss of generality, we consider $e = 1$. The coupling $g$ corresponds to the pairing contribution of the two-body interaction $\hat{V}$. Note that the RPH can be rewritten as (in the total spin $S = 0$ channel only):

$$\hat{H}_{RPH} = e \sum_{n\sigma}(n-1)\hat{a}_{n\sigma}^\dagger \hat{a}_{n\sigma} - \frac{g}{2}\sum_{nm}\hat{P}_n^+ \hat{P}_m^- , \tag{31a}$$

in terms of the pair creation/annihilation operators defined as:

$$\hat{P}_n^+ = \hat{a}_{n\uparrow}^\dagger \hat{a}_{n\downarrow}^\dagger , \quad \text{and} \quad \hat{P}_m^- = \hat{a}_{m\uparrow}\hat{a}_{m\downarrow} . \tag{31b}$$

---

[5]This is a simplified version of the RPH: in a more realistic case, $e$ and $g$ are not constants.

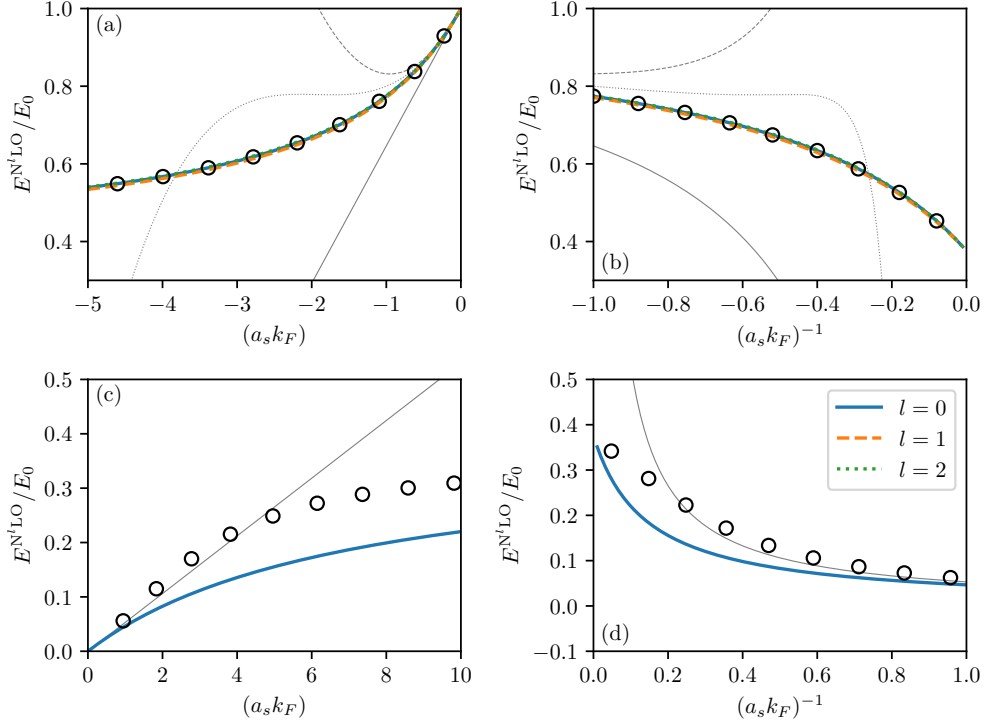

Figure 1: GS energy of the spin-saturated Fermi gas $E_\lambda^{N^l LO}$ discussed in section 4 as a function of $(a_s k_F)$ [panels (a) for $a_s < 0$ and (c) for $a_s > 0$] and $(a_s k_F)^{-1}$ [panels (b) for $a_s < 0$, and (d) for $a_s > 0$] obtained with (19) using the dilute Fermi gas correspondence (see section 4) for $l = 0$, $l = 1$, and $l = 2$. $E_0$ corresponds to the exact GS energy of the non-interacting Fermi gas. For reference, using thin gray lines, the first, second, and third orders of MBPT for the dilute Fermi gas [46] are shown, and, the *ab initio* calculations of [48, 49] are represented in each panel by black circles.

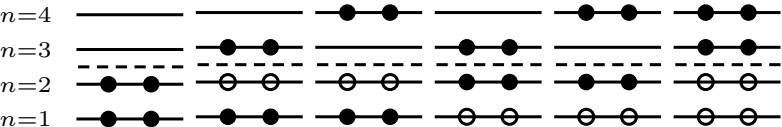

Figure 2: Configurations (Slater determinants) of the RPH discussed in the text with $N = 4$ and $p = 4$ ($N_p = 2p$). On the left is displayed the GS and zero-particle-zero-hole excitations ($0p - 0h$), the four central diagrams correspond to the $2p - 2h$ excitations, and the right diagrams correspond to the $4p - 4h$ excitations.

This Hamiltonian commutes with the product of the pair creation and annihilation operators and thus corresponds to a system with no broken pairs, i.e. the RPH link two-particle states in spin-reversed states.

In the following, we consider a system with $N = 4$ and no broken pairs (total spin $S = 0$) in the GS. We only consider $N_p = 8$ single-particle states, that is, $p = 4$ levels. The schematic configurations of this model are shown in figure 2.

Table 2: Energy of the single-particle states for the RPH discussed in the text labeled by the quantum number $n$ and the spin-projection on the quantification axis $\sigma$.

| $n$ | $\sigma$ | $E_{n\sigma}$ |
|---|---|---|
| 1 | $\pm 1/2$ | $-g/2$ |
| 2 | $\pm 1/2$ | $e - g/2$ |
| 3 | $\pm 1/2$ | $2e$ |
| 4 | $\pm 1/2$ | $3e$ |

Using table 2 and figure 2, we deduce that the Hamiltonian has the following matrix form:

$$\hat{H}_{RPH} = 2e \begin{bmatrix} 1 & 0 & 0 & 0 & 0 & 0 \\ 0 & 2 & 0 & 0 & 0 & 0 \\ 0 & 0 & 3 & 0 & 0 & 0 \\ 0 & 0 & 0 & 3 & 0 & 0 \\ 0 & 0 & 0 & 0 & 4 & 0 \\ 0 & 0 & 0 & 0 & 0 & 5 \end{bmatrix} - \frac{g}{2} \begin{bmatrix} 2 & 1 & 1 & 1 & 1 & 0 \\ 1 & 2 & 1 & 1 & 0 & 1 \\ 1 & 1 & 2 & 0 & 1 & 1 \\ 1 & 1 & 0 & 2 & 1 & 1 \\ 1 & 0 & 1 & 1 & 2 & 1 \\ 0 & 1 & 1 & 1 & 1 & 2 \end{bmatrix}. \tag{32}$$

### 5.1 Exact GS energy

In figure 3, we show the GS energy as a function of $g$. In the following, we introduce the *reduced* GS energy $\overline{E}$ defined as:

$$\frac{\overline{E}}{E_0} = \frac{E}{E_0} + \frac{3g}{2}\Theta(g), \tag{33}$$

where $E$ is the GS energy of the system and $\Theta$ is the Heaviside step function. This definition of the reduced GS energy allows for a finite limit in the high-scale limit, that is, $\lim_{g\to\pm\infty}\overline{E}/E_0 = 3$, which gives the high-scale constraint $\xi_0 = 3$.

### 5.2 Hartree-Fock and MBPT solutions

#### 5.2.1 Hartree-Fock

Starting in the single-particle basis $|i\rangle = |n_i\sigma_i\rangle$ given by table 2 (eigenvectors of $\hat{H}_0$), we define the single-particle HF basis as:

$$|i\rangle_{HF} = \sum_j \mathcal{C}_{ij}|j\rangle, \tag{34}$$

and then the HF equation reads (eigenvalues problem):

$$\sum_j h_{kj}^{HF}\mathcal{C}_{ij} = \epsilon_i^{HF}\mathcal{C}_{ik}, \tag{35a}$$

where the HF matrix is defined as:

$$h_{ij}^{HF} = \langle i|\hat{h}_0|j\rangle + \sum_{k=1}^{N}\sum_{pq}\langle ip|\hat{v}|jq\rangle, \tag{35b}$$

with $\langle i|\hat{h}_0|j\rangle = e\delta_{ij}$ and:

$$\langle ip|\hat{v}|jq\rangle = -\frac{g}{2}(-1)^{\sigma_i+\sigma_j}\delta_{n_i n_p}\delta_{n_j n_q}(1-\delta_{\sigma_i\sigma_p})(1-\delta_{\sigma_i\sigma_q}), \tag{35c}$$

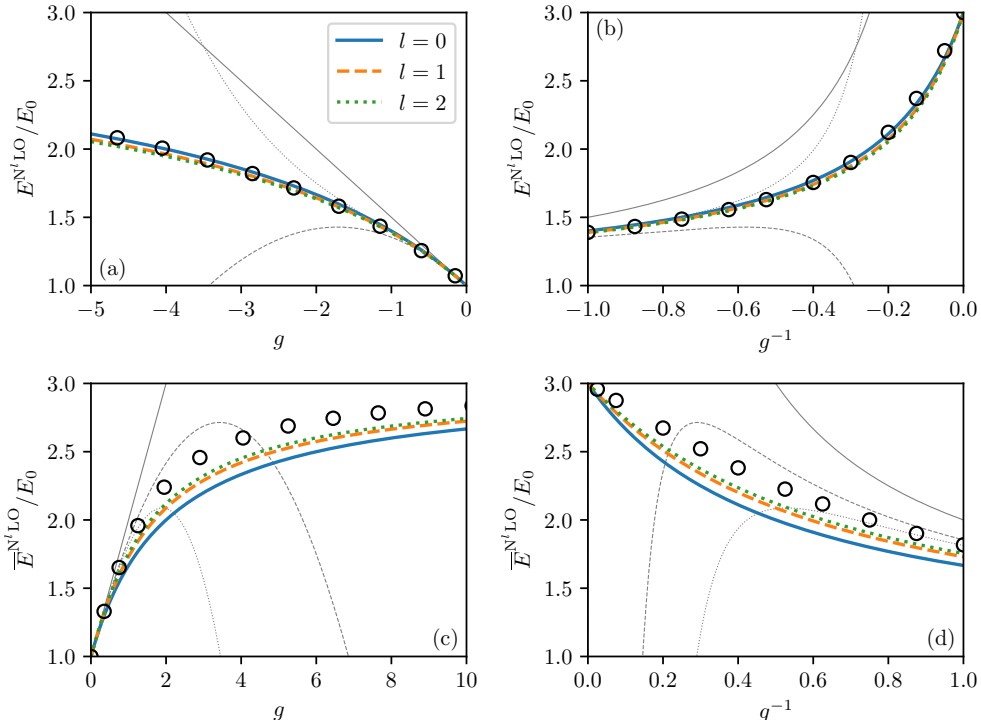

Figure 3: GS energy $E_\lambda^{N^l LO}$ or reduced GS energy $\overline{E}_\lambda^{N^l LO}$ discussed in the text for the RPH as a function of $g$ [panels (a) for $g < 0$ and (c) for $g > 0$] and $g^{-1}$ [panels (b) for $g < 0$, and (d) for $g > 0$] obtained with (19) using the low-scale parameter correspondence of the model for $l = 0$, $l = 1$, and $l = 2$. $E_0$ corresponds to the reference energy of the GS. For reference, using thin gray lines, the first, second, and third orders of MBPT are shown, and the exact result is represented in each panel by black circles.

is antisymmetrized under the exchange of particles.

We can show that the solution to the HF equation is given by $\mathcal{C}_{ij} = \delta_{ij}$ and we deduce the HF GS energy:

$$E_{HF} = \sum_{i=1}^{N} \sum_{j=1}^{N} \left[ \langle i|\hat{h}_0|j\rangle - \frac{1}{2} \langle ij|\hat{v}|ij\rangle \right] = 2e - g. \tag{36}$$

We note that the HF GS energy is equal to the reference energy obtained considering only the $0p - 0h$ excitation, i.e. $E_{HF} = E_{0p-0h}$.

### 5.2.2 MBPT results

Calculations of the many-body perturbation theory provide the low-scale expansion of the GS energy for the pairing model discussed in this section, i.e. the coefficients $\{\gamma_n\}$ appearing in (2) up to fifth order are given by: $\gamma_1 = -1/2$, $\gamma_2 = -7/48$, $\gamma_3 = -1/24$, $\gamma_4 = -77/27648$, $\gamma_5 = 5/864$, etc. For the reduced GS energy (33), only the first order is affected and $\overline{\gamma}_1 = \gamma_1 + 3\Theta(g)/2$ and $\overline{\gamma}_n = \gamma_n$ where $\{\overline{\gamma}_n\}$ are the coefficients of the low-scale expansion (2) for the reduced GS energy $\overline{E}$. These results are summarized in table 3.

In practice, we perform the method proposed in this paper for the reduced GS energy $\overline{E}$ using the low-scale coefficients $\overline{\gamma}_n$ and the high-scale constraint $\xi_0 = 3$, then the energy is obtained simply using (33), i.e. $E^{N^l LO} = \overline{E}^{N^l LO} - 3gE_0\Theta(g)/2$.

Table 3: Low-scale coefficients and high-scale limit, for $g > 0$ and $g < 0$, of the RPH model introduced in the text. In parentheses, we give the values of the regularized coefficients to remove divergences of the GS energy when it is suitable.

|            | $g > 0$        | $g < 0$  |
| ---------- | -------------- | -------- |
| $\gamma_1$ | $-1/2$ (1)     | $-1/2$   |
| $\gamma_2$ | $-7/48$        | $-7/48$  |
| $\gamma_3$ | $-1/24$        | $-1/24$  |
| $\xi_0$    | $-\infty$ (3)  | 3        |

The final result for the GS energy as a function of the coupling constant $g$ is shown in figure 3. We observe that, even at leading order, the GS energy is reproduced in good approximation in a wide range of interaction strengths. In particular, contrary to the standard MBPT results that diverge for $|g| > 1$ (thin gray lines), our approaches grasp most of the BMF correlations in the strong coupling regime.

# 6 Condensed matter physics context: Application to the Hubbard Hamiltonian

In this section, we apply our strategy developed above for the description of a model that describes low-energy physical properties of strongly correlated fermions navigating into a lattice, namely the Hubbard model.

The Hubbard model [51–53] stands as one of the simplest and most frequently used effective models in theoretical condensed matter physics [54,55]. Its purpose is to capture the general properties of spin-1/2 electrons moving through a lattice by hopping between neighboring sites and subject to a local two-body interaction with strength $U$. In its second-quantized form, the correlated tight-binding Hamiltonian is expressed as:

$$\hat{H}_{\text{Hubbard}} = -t \sum_{\langle i,j \rangle, \sigma} \left( \hat{c}^\dagger_{i\sigma} \hat{c}_{j\sigma} + \hat{c}^\dagger_{j\sigma} \hat{c}_{i\sigma} \right) + U \sum_i \hat{n}_{i\uparrow} \hat{n}_{i\downarrow}, \tag{37}$$

where $t$ is the hopping integral, $\hat{c}^\dagger_{i\sigma}$ and $\hat{c}_{i\sigma}$ are, respectively, the creation and annihilation operators of electron with spin $\sigma = \uparrow, \downarrow$ on site $i$, the $\hat{n}_{i\sigma} = \hat{c}^\dagger_{i\sigma} \hat{c}_{i\sigma}$ is the occupation number operator, and $U$ is the on-site nearest-neighbor Coulomb repulsion of spin-up and spin-down electrons occupying the same lattice site. We note that in the positive $U$ regime, the model received a considerable renewed interest in two-dimensional (2D) geometry after P. W. Anderson's proposal in connection to high-$T_c$ superconducting cuprates [56,57]. Furthermore, the one-dimensional (1D) Hubbard model [58] has previously also been proposed as a minimal model to describe the low-energy physical properties of 1D conductors [59] and quasi-1D copper oxides compounds [60–62]. In the following, we will put $t = 1$ to set our unit of energy without losing generality. Note that the summation appearing in the kinetic term in (37) is done on the nearest neighbor sites $j$ for all sites $i$ designed by $\langle i, j \rangle$.

For the sake of simplicity of the discussion and to be able to compare our results with the exact solution, we restrict the model to the so-called half-filled 1D Hubbard chain and to the half-filled 2D four-sites Hubbard Hamiltonian for which exact formulas of the GS energy from exact diagonalization can be found in the literature [63,65].

## 6.1 Application to the 1D Hubbard chain model

As mentioned above, we apply our strategy to the 1D Hubbard chain model described by the Hubbard Hamiltonian (37) at half-filling and considering only nearest-neighbor hopping amplitudes with a positive on-site repulsion ($U > 0$).

The exact GS energy is then obtained using the Bethe anzats and is given in [63]:

$$\frac{E}{E_0} = \pi \int_0^\infty \frac{d\omega}{\omega} \frac{J_0(\omega) J_1(\omega)}{1 + \exp((U/t)\omega/2)}, \tag{38}$$

where the $J_n$ are the Bessel functions of the first kind. This expression can then be expanded to the low-scale limit $U/t \ll 1$ leading to the low-scale coefficients of expansion (2) and presented in table 4 [64]. Taking the high-scale limit provides the high-scale parameter $\xi_0 = 0$ for $U/t > 0$ and $\xi_0 = 2$ for $U/t < 0$.

The results of the strategy developed in this article and applied in the previous sections, are given in figure 4. Again, we observe a rather good approximation of the GS energy along the wide range of $(U/t)$ values. We mention that the strategy is not applicable at the third order because of the vanishing of the third order term in the low-scale expansion ($\gamma_3 = 0$). However, we argue that careful decomposition of the fourth-order MBPT expansion will lead to a substantial improvement. We stick here to the third order to be consistent with the level of approximation employed in the discussion above.

## 6.2 Application to the four-sites 2D Hubbard model

Finally, we follow our strategy for the four-sites 2D Hubbard model (square $2 \times 2$ cluster) that also takes the prerequisites decomposition (37) for which the exact GS energy can be found in [65] and takes the expression:

$$\frac{E}{E_0} = -\frac{U}{4t} + \frac{1}{2}\sqrt{\frac{16 + (U/t)^2}{3}} \cos\left(\frac{\theta}{3}\right), \tag{39}$$

where:

$$\theta = \arccos\left(\frac{4U}{t}\left(\frac{3}{16 + (U/t)^2}\right)^{3/2}\right),$$

leading to the low- and high- scale parameters given in table 5. The only subtlety is that, in this case, we have a linear divergence of the GS energy for $(U/t) < 0$ that can be treated as the one appearing in the study of the Richardson Pairing Hamiltonian in section 5. The results are compared to the exact calculations and MBPT results in figure 5 for the first, second, and third order expansion. We observe that the GS energy is reproduced accurately at the third order in perturbation. Also, convergence according to the level of approximation is obtained, as expected by the formal aspect of our expansion.

Table 4: Low-scale coefficients and high-scale limit, for $(U/t) > 0$ and $(U/t) < 0$, of the 1D Hubbard chain model introduced in the text.

|  | $(U/t) > 0$ | $(U/t) < 0$ |
|---|---|---|
| $\gamma_1$ | $-\pi/16$ | $-\pi/16$ |
| $\gamma_2$ | $7\zeta(3)/64\pi^2$ | $7\zeta(3)/64\pi^2$ |
| $\gamma_3$ | 0 | 0 |
| $\xi_0$ | 0 | 2 |

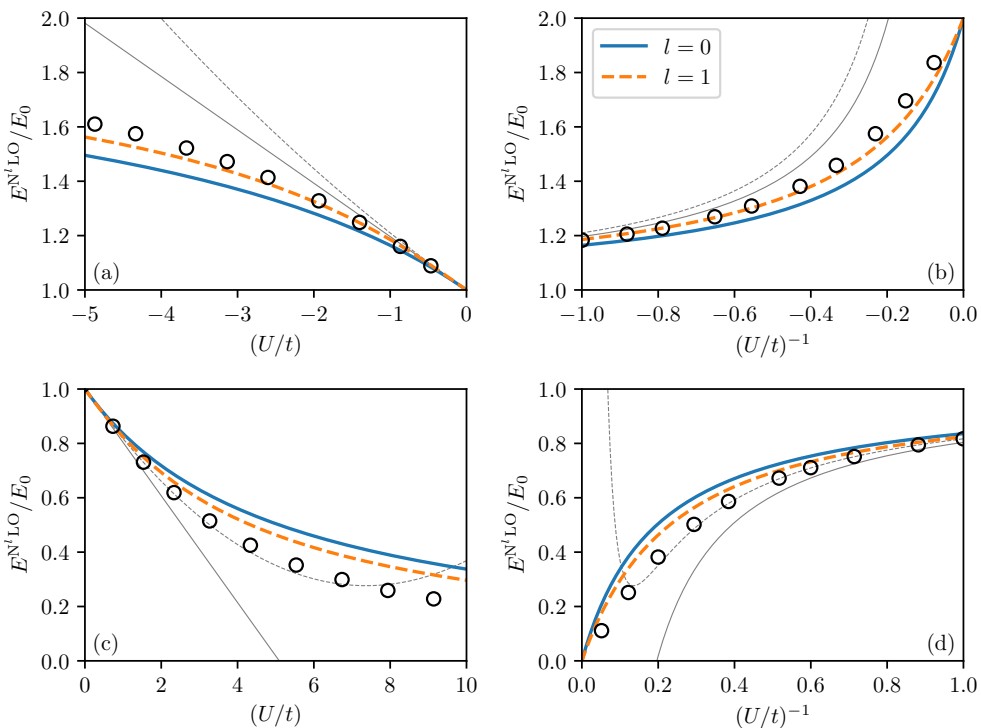

Figure 4: GS energy $E^{N^l LO}_\lambda$ discussed in the text for the 1D Hubbard chain model as a function of $U/t$ [panels (a) for $U/t < 0$ and (c) for $U/t > 0$] and $(U/t)^{-1}$ [panels (b) for $U/t < 0$ and (d) for $U/t > 0$] obtained with (19) using the low-scale parameter correspondence of the model for $l = 0$ and $l = 1$. $E_0$ corresponds to the reference GS energy. For reference, using thin gray lines, the first and second orders of MBPT are shown, and the exact result is represented in each panel by the black circles.

Table 5: Low-scale coefficients and high-scale limit, for $(U/t) > 0$ and $(U/t) < 0$, of the four-sites 2D Hubbard model introduced in the text. In parentheses, we give the values of the regularized coefficients to remove divergences of the GS energy when it is suitable.

|  | $(U/t) > 0$ | $(U/t) < 0$ |
|---|---|---|
| $\gamma_1$ | $-3/16$ | $-3/16\ (5/16)$ |
| $\gamma_2$ | $13/512$ | $13/512$ |
| $\gamma_3$ | $-3/1024$ | $-3/1024$ |
| $\xi_0$ | $0$ | $\infty\ (0)$ |

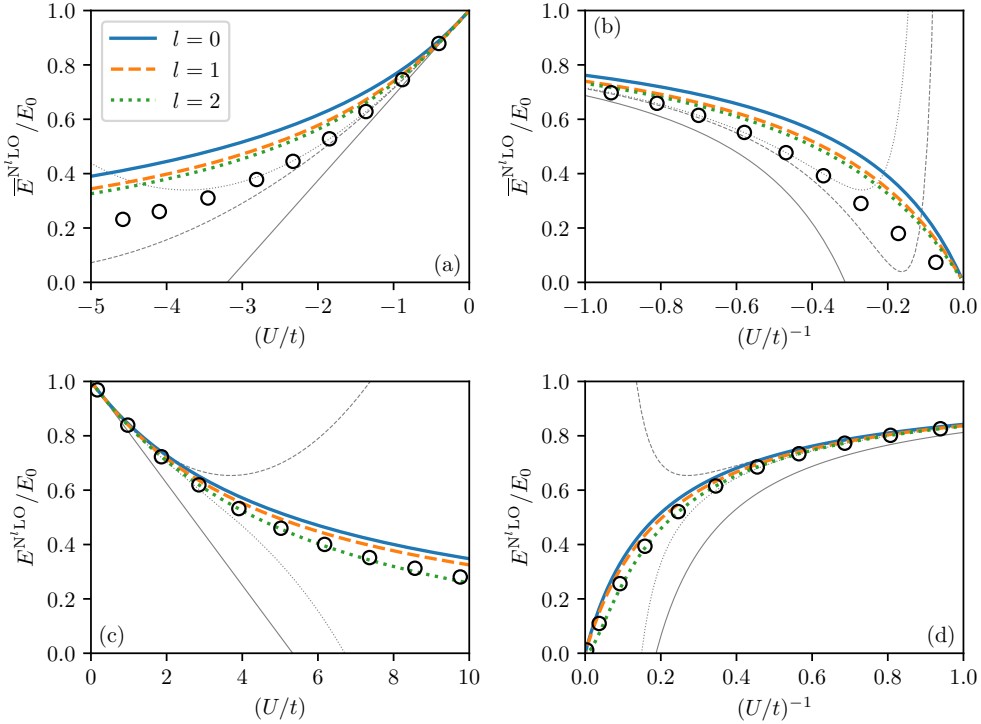

Figure 5: GS energy $E_\lambda^{N^l LO}$ or reduced GS energy $\overline{E}_\lambda^{N^l LO}$ discussed in the text for the four-sites Hubbard model as a function of $U/t$ [panels (a) for $U/t < 0$, and (c) for $U/t > 0$] and $(U/t)^{-1}$ [panels (b) for $U/t < 0$, and (d) for $U/t > 0$] obtained with (19) using the low-scale parameter correspondence of the model for $l = 0$, $l = 1$, and $l = 2$. $E_0$ corresponds to the reference GS energy. For reference, using thin gray lines, the first, second, and third orders of the MBPT are shown, and the exact result is represented in each panel by the black circles.

## 7  Conclusion: Discussion and outlook

In this work, we emphasized the possibility of exhibiting free parameters in the MBPT expansion that allow for the use of effective Hamiltonians that contain BMF correlations. In particular, we show that the problem of double counting of BMF correlations, arising from the use of many-body techniques on a renormalized theory, can be automatically solved due to the flexibility of the expansion without fitting procedure or further adjustments. The main advantage of the strategy used in this paper is twofold:

- A systematic convergence of the results, faster than the standard MBPT, is observed at each order of truncation without inducing an increasing computational complexity.

- An explicit parameterization of the GS energy (and of the many-body states) is obtained in terms of the MBPT amplitude and high-scale limit of the bare theory, that is to say, a finite and restricted number of physical constants, e.g. $\{\gamma_n\}$, accessible from standard many-body theory or analysis of experimental data.

We can mention some restrictions since the applicability of our strategy depends strongly on the state-of-the-art standard MBPT, *ab initio* methods, and experiments to provide the low-scale constants and the high-scale limit of the bare Hamiltonian.

A compelling aspect resides in the fact that the formalism is independent of the parameterization of the correlated effective Hamiltonian chosen. As shown in this work, our extended MBPT is valid for various correlated systems, from ultracold atoms and finite nuclei, to condensed matter systems described by the Hubbard Hamiltonian. The simple cases for which we tested our strategy have highlighted some restrictions that require further investigations to grasp most of the complexity of realistic models [66–78]. We contend that our approach to characterize quantum many-body problems possesses a degree of universality, as it allows the integration of parameters that can be adjusted to match specific physical properties.

This novel approach could finally be a valuable guide providing new insights and ideas to predict the collective behavior of quantum many-body systems, e.g. via the linear response theory [79–82] that requires the knowledge of the ground state. Another aspect of the new method developed in this paper is the possibility of designing a new parameterization of BMF DFT that extends the domain of applicability for a wide range of systems. This is also a promising way to link DFT and EFT [39, 83]. Finally, due to the link between MBPT and Coupled-Cluster theory, it can be interesting to investigate possibilities to apply our method in such a formalism.

## Acknowledgments

We warmly thank Philippe Cirot, the former director of our institute, who unconditionally and systematically encouraged our work.

**Conflict of interest** The authors declare that they have no conflict of interest.

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
