# Peer review of "Many-body perturbation theory for strongly correlated effective Hamiltonians using effective field theory methods"

_SciPost Physics, doi:SciPost Phys. 18, 003 (2025)_

## Round 1 · Referee Report · Anonymous (Referee 1) · 2024-4-16

Strengths

See report.

Weaknesses

See report.

Report

The authors propose a many-body perturbation theory for strongly correlated systems using effective field theory methods. The main idea is to start from an effective Hamiltonian which allows one to reproduce the perturbation expansion in some interaction parameter while recovering exactly the known (exact) result in some (strongly-correlated) limit. The goal is to set up an approach which can interpolate between the weak- and strong-correlation limits. The main problem is to avoid double counting of correlations when going beyond lowest-order perturbative expansion.

The paper is clearly written. I would like to mention the following points for the authors' consideration.

1) In the introduction, the authors point out that "non-perturbative methods express the problem with multidimensional integrals". It is not clear what is referred to here. Slightly below, they identify first-order perturbation theory with mean-field theory, which I find slightly confusing. Can BCS mean-field theory be considered as a first-order perturbation theory?

2) The main point of the authors is to show that, starting from Hamiltonian (4), one can reproduce the perturbation expansion order by order while satisfying the exact result $E_\infty/E_0=\xi_0$. To do so, one has to introduce an unknown parameter, $\beta$, to second order; two parameters $\beta_1$ and $\beta_2$ to third order order, etc. The procedure followed to second order, Eq.(26), seems rather arbitrary. Could the authors justify it? Is it the only possible way to introduce the two parameters $\beta_1$ and $\beta_2$ and, if not, why choosing this one?

3) I do not understand the meaning of the sentence "which is again independent of $\beta_1$ and $\beta_2$" following Eq.(26).

4) It is shown how to reproduce the perturbation expansion order by order. I understand, although it is not said explicitly, that this is equivalent to avoiding double counting of correlations. A short discussion would be welcome.

5) At the top of page 4, "an energy operator $\hat\omega|\Psi_0\rangle$" should be replaced by "an energy operator $\hat\omega$".

6) The various examples considered in the manuscript are quite convincing except the 1D Hubbard model. In the case $U/t>0$, it seems that the second-order perturbation theory results are better than the $l=0$ and $l=1$ results. Moreover I do not understand what the model with $l=2$ and $l=3$, mentioned in the caption of Fig.4, refer to.

7) The authors discuss only the calculation of the ground state energy. In many-body systems, correlation functions are also of prime interest. Is the method proposed in the manuscript restricted to thermodynamic quantities or would it be possible to also compute one- and two-particle Green functions?

Requested changes

See report.

Recommendation

Ask for minor revision

  • validity: good
  • significance: good
  • originality: good
  • clarity: good
  • formatting: good
  • grammar: good

Author:  Antoine Boulet  on 2024-05-17  [id 4491]

(in reply to Report 1 on 2024-04-16)

Dear Editor, Dear Reviewers,

We would like to thank you for your time in reviewing our paper and providing valuable comments that led to possible improvements in the current version. We have carefully considered the comments and tried our best to address every one of them. We hope that the manuscript after careful revisions, will meet your high standards. We welcome further constructive comments if any. In the file attachment, we provide the point-by-point responses.

Sincerely,

R. Photopoulos and A. Boulet

Attachment:

responsev1.pdf

---

## Round 2 · Referee Report · Anonymous (Referee 1) · 2024-7-16

Report

The authors have responded satisfactorily to my comments and questions and I now recommend the manuscript for publication.

Recommendation

Publish (meets expectations and criteria for this Journal)

---

## Round 2 · Referee Report · Anonymous (Referee 2) · 2024-10-5

Strengths

See Report

Weaknesses

see Report

Report

Report on
“Many-body perturbation theory for strongly correlated effective Hamiltonians using effective field theory methods”
by Raphaël Photopoulos and Antoine Boulet

The authors, R. Photopoulos and A. Boulet develop a strategy for applying many-body perturbation theory starting from an effective Hamiltonian that already contains correlations beyond the mean-field level. In particular, their method provides a solution to the over-counting of correlations that is known to be inherent in this type of approach.
Overall, this work is clear, well presented and well written. Several examples are presented and compared to exact results in order to illustrate the validity and accuracy of their theoretical method.

I would now like to make a few comments on specific points raised in the manuscript.

1- In order to discuss the validity of their theoretical approach, the authors focus essentially on the ground-state energy as a function of the relevant physical parameter of the many-body Hamiltonian under consideration. However, in several illustrative models considered in their study, other relevant quantities such as correlation functions could be calculated. This would allow a better assessment of how close the N-body ground-state calculated in their approach is to the exact many-body ground state. It is as well possible to calculate directly the overlap between the exact ground state and the one they calculate in their MBPT approach. For instance, this could be achieved relatively easily in the case of the four-site Hubbard model and even in the case of the Richardson pairing Hamiltonian.

2- In Fig.4, which concerns the case of the one-dimensional Hubbard chain, the calculations corresponding to l=2 (third order perturbation) are not shown, why? The authors should present the results the agreement should be better than for l=1?

3- In the case of Hubbard's four-site model, it would appear, in the attractive case, that agreement decreases as the order of perturbation increases. For example, the agreement between the exact calculations and the MBPT calculations for l=0 is excellent, whereas as the order of perturbation increases, it decreases. Do the authors have an explanation?

4- In Figures 1, 4 and 5, the left and right panels (a) and (b) are the same data plotted as a function of the relevant parameter or its inverse. In my opinion, the authors should choose one of them. There's no need to keep both, there's no advantage in doing so, it doesn't help to better understand their results and the comparison between the exact calculations and MBPT calculations.

Recommendation

Ask for minor revision

  • validity: good
  • significance: high
  • originality: good
  • clarity: high
  • formatting: excellent
  • grammar: excellent

Author:  Antoine Boulet  on 2024-11-04  [id 4931]

(in reply to Report 2 on 2024-10-05)

Dear Editor, Dear Reviewers,

We would like to thank you for your time in reviewing our paper and providing valuable comments that led to possible improvements in the current version. We have carefully considered the comments and tried our best to address every one of them. We hope that the manuscript after careful revisions, will meet your high standards. We welcome further constructive comments if any. In attachment, we provide the point-by-point responses.

Sincerely,

R. Photopoulos and A. Boulet

Attachment:

response_2.pdf

---

## Editorial Decision

published